# An Investigation of Preschool Level Out-of-Class Education Activities in Finland, Estonia, Ireland, and Turkey within the Framework of 21st Century Skills

Kezban Özgem [1,*] and Umut Akçıl [2]

1   Faculty of Education, Near East University, Nicosia 99138, Cyprus
2   Educational Science Department, Faculty of Education, Near East University, Nicosia 99138, Cyprus; u.akcil@neu.edu.tr
*   Correspondence: kezbanozgem@gmail.com; Tel.: +90-533-830-1034

**Abstract:** The skills and competencies to be acquired in the 21st century are collected under these topics: learning and innovation skills; life and career skills; information, media and technology skills. The changes taking place in the world, in information technologies and global transformation, have promoted the development of different approaches, models, skills, and various learning theories in education. One of the prominent features shaping this period is the acquisition of these desired skills from the pre-school period, and this acquisition occurs through providing children with different experiences and offering rich environments and materials to children. In this respect, the activities carried outdoors have gained as much significance as in-class activities. This study aims to compare the 21st-century skills-based outdoor educational activities in Finland, Estonia, Ireland, and Turkey. In this respect, the horizontal and diagnostic approach used in comparative education studies is applied in combination. Document analysis is used in this research to gather data on the objectives of the countries regarding preschool out-of-class education as well as out-of-class education environments; the activities, modules, and assessment methods used were obtained from Finland, Estonia, Ireland, and Turkey's Ministry of Education websites, countries' laws on education, official pre-school education reports, education systems, articles, and online databases, etc. Among the countries examined, Finland, Estonia, and Ireland have learning modules related to out-of-class learning activities, which indicates that these countries have more options and thus a greater variety of out-of-class activities and environments. It also indicates that out-of-class education activities are carried out more extensively in these countries. It has been concluded that Ireland has more out-of-class learning activities in number and a number of the activities address a higher number of developmental areas. However, it has also been revealed that there is no module, program, or booklet for out-of-class learning activities in Turkey. From this point of view, the recommendation is for Turkish education to create a preschool education program module with out-of-class activities to adapt the Turkish preschool education program to the imperatives of contemporary learning outcomes.

**Keywords:** 21st century skills; pre-school education; out-of-class education; activity; comparative

## 1. Introduction

Today's technological developments have brought about changes in the needs of individuals and altered the understanding of education and our perspective on it. This calls into question the determination of the skills required to keep up with the changing and developing world, to follow the innovations and to adjust to these changes as well as including the applications that have been used during this transformation from the past to the present [1]. These practices are called 21st-century skills [2]. With these changes, the important feature is not for students to receive the information directly, but to acquire the skills to find the information correctly and effectively to transfer the information in a healthy way [3]. It does not seem possible to acquire 21st-century skills such as creativity,

critical thinking, problem-solving, communication, technology literacy, social skills, and collaborative work with the understanding of classical education in classrooms. Therefore, the use of different learning environments is necessary [4]. One of the most critical points emphasized in 21st-century skills is that they are not only knowledge and skill-oriented, but also value understanding and are performance-oriented [5]. Various institutions have classified 21st-century skills differently [6]; ATCS (Assessment and Teaching of 21st Century Skills), P21 (Partnership for 21st Century Learning), OECD (Organization for Economic Co-operation and Development), ASIA Society (Asia Society Partnership for Global Learning), ISTE (International Society for Technology in Education), NCREL (North Central Regional Educational Laboratory), EU (European Union) have classified these skills in different ways. Despite the fact that there are various classifications of these skills, they possess some common features. In the frameworks of 21st-century skills, it was emphasized that prominent individuals should be able to think creatively, be able to hold a critical vision, evaluate situations they encounter, cooperate with various other groups, and solve the encountered problems. 21st-century skills emphasize the importance of not necessarily having ready-made knowledge; rather, they underscore the importance of having the competence to reach the information and to use the information creatively and effectively. It also includes individuals being able to live together with different cultures and respect them. Being an active, productive citizen is important an important learning outcome in the 21st century. Given the importance of technology in our lives, possessing the ability to use these tools effectively and being multiliterate in literacies (information, media, digital age) are crucial skills. People endowed with these skills become more qualified and productive, so as it is necessary to include these skills in education programs, and it has become necessary for students to acquire these skills through education [2]. Trying to teach scientific facts only in the classroom or laboratory setting is not sufficient for the multifaceted development of the students. This led to an increase in the current interest in outdoor education concept. Outdoor education emerged primarily for environmental education purposes [7]. Outdoor education includes the processes done 'outside of school', parallel to the education program, and supports purposeful learning. At this point, out-of-school learning activities (outdoor learning activities) have started to be intertwined with learning areas such as place-based learning, experiential learning, and learning in a real-life context by being associated with external learning over time [8]. The concept of outdoor education is used in different ways: education in nature, environmental education, outdoor education, out-of-class education or out-of-school education, experiential education, adventure education, etc. [9–11]. The study of the relevant literature suggests shows that there is no consensus on how to define the concept of outdoor education [12]. These varied uses are due to the various purposes for which and the locations in which educational activities are carried out. In this context, when we look at outdoor education activities, it is seen that the activities are carried out not only in open areas, but also in closed places such as factory tours, swimming pools, and science centers [13,14]. In addition, outdoor education are often limited by a narrow emphasis on the environment. However, the outdoors could be an asset not only in teaching students about the environment; they could also be used in different subjects such as communication, mathematics, and history, geology [15].

Out-of-class education activities conducted from early childhood onwards go beyond the traditional lesson-oriented approach and ensure active learning. The chosen student-oriented approaches ensure their active participation, and enable the students to have a voice by placing them at the center. Outdoor activities provide students with socio-emotional support along with academic knowledge. The stimulation of children's natural curiosity encourages students to be interested in scientific facts, and supports students with inquiry-based experiments and learning experiences [16]. Play occupies a crucial place during childhood and constitute the basic elements of learning. Children learn while having fun playing games. Their curiosity becomes activated. Learning becomes meaningful and interesting since they are active in the activation process. It supports children's learning and development. Games provide fun learning opportunities for children. The crucial

thing is to set goals and choose appropriate activities. Different areas of development can be supported through the selection of different games [17].

Open-air education is gaining more importance day by day; the importance of nature on learning is becoming clear. Open space activities, as an innovative pedagogical approach, provide numerous benefits to students. Although the locations of schools and their environments create different teacher perceptions on this issue, and possess differences on the outdoor educational activities, its benefits for students are obvious [18]. In early childhood, learning and development takes place through encountering the real world and exploring through social interaction. Gardens are the perfect setting to offer this opportunity [19]. It is vital to understand and learn the biodiversity of a sustainable environment. Biodiversity education in the early childhood period starts with observations of plants and animals in nature. Therefore, it is crucial for students to become acquainted with nature and learn the local nature in the early childhood period [20]. Educational environments affect children's participation rates in activities. In their research, Clevenger & Pfeiffer (2022) revealed that the garden, lawn area, sandbox area, and areas in which objects are used are the areas where children are most active [21]. Out-of-school learning activities carry the meanings of learning, contribution, reinforcement of learning, and environment. It is argued that outdoor education environments enable children to learn permanently, reinforce their learning and gain experience, and develop skills [22]. In early childhood, activities carried out outdoors and in contact with nature show that they offer benefits related to the health and development of children. Environmental awareness, interdisciplinary learning, and the effectiveness of nature in gaining various academic subject skills are emphasized [23].

It is stated that life skills should be given more importance in education, distinct from theoretical, technical knowledge or skills already offered in schools in order to raise individuals who can cope with changes which is a more critical requirement from the past [24]. The 21st century is accepted as the beginning of the digital age with the rapid growth of technology and the subsequent information explosion. In the 21st century, one of the most essential conceptual shifts in education has been the experience of digital transformation. One of the main aims of the concept of education has been to enable the individual to access information, instead of transferring information. In line with this goal, education gained a new definition by combining curiosity, questioning, and research skills. In this context, educational environments, and practices have changed in recent years with 21st-century learning and gained renewal skills of creativity, renewal, problem-solving, decision making, critical thinking, communication, and cooperation [25].

21st-century skills are the competencies that today's individuals need to acquire. In Ireland, the framework developed covers students aged 3 to 16 years, with the aim to integrate these skills into the curriculum from an early age. Coding and programming courses have been made compulsory for students between the ages of 5 and 16 in Ireland, and efforts are being made to spread digital literacy not only to these courses but also to the entire curriculum. Likewise, in Ireland, the importance of teaching these skills in all their aspects and developing general skills and competencies in the curriculum are emphasized. With the new national curriculum, digital technologies, digital literacy, and communication are expected to be covered in all education levels, all subjects, and cross-curricular topics [26].

At all levels of education, educational activities taking place inside and outside the classroom necessitates interactivity. It is an evident fact that learning and renewal skills and behaviors are not only limited to in-school activities but are similarly dependent on the effectiveness of outdoor learning environments that support learning experiences [26–28]. Outdoor education is one of the most effective methods or strategies used to enable students to comprehend curriculum-related practices that are difficult or impossible to convey in the classroom [29]. Outdoor learning is not limited to a field trip or an outdoor lesson. In the education, teaching processes, classroom, outdoor practices, and activities include sightseeing-observation; terrain-field studies; trips or visits to social, cultural and scientific environments (museums, natural history museums, science and technology museums,

botanical gardens, zoos, planetariums, meteorology stations, water treatment plants, dams, industry) organizations, official institutions, educational institutions. Teaching practices also involve virtual reality applications; nature and environmental education; environmental club activities; homework, projects related to places; sports activities; social, cultural, and scientific programs; and spatial applications for lifelong learning. These are among the activities that have gained importance in recent years. Such teaching is associated with a wide variety of learning contexts, such as Children's Universities opened within universities and TÜBİTAK-supported science and society projects (such as nature education and science schools, science festivals, and science fairs), as well as environments that support outdoor learning environments [30].

Based on the literature review, the acquisition of 21st Century competencies defined by the OECD, Finland, Estonia, and Ireland are seen as a result of digitalization and a condition to reach sustainability. The approaches mentioned used in the European continent on this subject, Finland, and Estonia are in the top five in the 2019 media literacy index, which is the most up-to-date index among European countries. The emphasis Ireland places on gaining these skills in their curricula has led to the selection of these countries. The reason Turkey is also included in this comparative study is that the researchers are based in Turkey, and the topic of research is relevant to pedagogy in Turkey [31,32].

Outdoor education contributes to the development of children's social aspects, awareness of the concept of time and managing time, strengthening friendship relations, self-development, self-confidence, leadership spirit, and emotional awareness [11,33–36]. Accordingly, Wagner [37], states in his study that, in order to be successful in the global economy, individuals must be taught different skills; otherwise, the competitiveness of these countries in the global economic race will worsen.

Outdoor education includes research and inquiry-based learning approaches. Students take an active role in the process in line with their curiosity and interests. Studies suggest students research, take responsibility, wonder, get interested in, ask questions, experiment, solve problems and construct their knowledge [38]. Indeed, Dillon [39], stated that out-of-class activities carried out in their education life were remembered by students for many years. Of course, remembering activities may not be considered an indicator of learning. The main purpose of outdoor education activities is effective learning and lasting outcomes. Therefore, such activities can also be used to reinforce learning activities carried out at school. For this reason, outdoor learning activities have positive effects on attitudes, values, and beliefs, as well as being entertaining, intriguing, and occupying an important place in the lives of individuals for a long time [40].

The evaluation of outdoor learning environments for students suggests that the school environments are considered to provide more teacher-centered learning, while out-of-class areas allow more student-centered learning. In another study, the effects of outdoor learning environments on lifelong learning skills were considered. The study determined that students' participation in outdoor learning environments, and their learning skills in these environments also affect their participation in educational activities in their professional lives [41].

The period between ages 3–6 is a significant period that covers an important place in childhood development, and this period ends when children reach the age of 6. It has become one of the important responsibilities for families, teachers, and educational institutions to provide environments and different materials that support the developmental areas of children qualitatively and productively during the pre-school period; if the desired outcome is for students to gain 21st-century skills, then mental transformation must first be carried out in the entire education system. To that end, Nieven [42] state that the acquisition of these skills is possible with studies to be carried out at the classroom, school level, and system level.

In Finland, early childhood education is the responsibility of parents in coordination with local governments. Early childhood education takes place with activities held in preschool education, daycare, and family daycare homes in kindergartens, and activities

held outdoors or in suitable places. Pre-school education is provided free of charge in Finnish, Swedish, and Sami the year before the start of compulsory education. The main goal of the program is to improve the child's growth, development, and learning conditions in cooperation with the family, and to strengthen children's positive self-perception by improving their social skills through play and positive learning experiences [43].

The structure of the education system in Estonia provides opportunities for everyone to move from one education level to another. Education levels include pre-primary education (ISCED level 0). The obligation to attend school applies to children who have turned 7 years old by 1 October of the current year. Children up to the age of 7 can attend pre-school child care institutions. The compulsory school attendance continues until basic education is received or the student is 17 years old. The Estonian education system is decentralized. The distribution of responsibilities between state, local governments, and schools is well defined. Local governments govern pre-school child care institutions. The education taking place in preschool child care institutions, schools create their own curricula when it comes to the curriculum—under a uniform national curriculum. The language of instruction is mainly Estonian, but other languages may also be used as specified in the legislation [44].

In Ireland, the education for the provision of Early Childhood Care and Education (ECCE) is delivered by teachers trained in infant classes in primary schools. The compulsory school age in Ireland is 6 and all forms of pre-primary education are optional. However, from the age of 4, children can be enrolled in infancy classes in primary schools. A 'free' Preschool Year program (Early Childhood Care and Education or ECCE programme) is administered by the Department of Child and Youth Affairs (DCYA) [45].

Turkey Early Childhood Education covers Nursery and Day Care Centers for children aged between 0–36 months old, operating under the General Directorate of Child Services of the Ministry of Family, Labor and Social Services. In addition to this, special education kindergartens, which are pre-school education centers for children 0–36 months old in need of special education, operate under the General Directorate of Special Education and Guidance Services. Children aged from 57–68 months enroll in kindergarten, nursery, and practice classes as of the end of September of the year of enrollment. After the registration of children residing in the registration area of the school and starting primary education in the next academic year, 36–56-month-old children can register to kindergarten and practice classes, and 45–56-month-old children can register to kindergartens in sufficient numbers. Pre-school education is carried out under the responsibility of the Ministry of National Education General Directorate of Basic Education [46].

In this research, the aim is to comparatively determine the outdoor education activities of Estonia, Finland, and Ireland, which are the best three countries among European countries in the field of reading, mathematics, and science according to the Pisa 2018 results, and to compare the findings with the outdoor education activities of Turkey. The fact that the three countries to be compared in the research have the best ranking among the European countries in the Pisa ranking, the presence of outdoor education modules in addition to preschool education programs, and the fact that they are among the countries with the highest comparison in the literature made them suitable for comparison in this study.

Comparative education emerges as an important area of discipline that deals with providing solutions and improving educational results to be more effective and efficient [47]. With these aspects, comparative education is an important branch of science that directs education and sheds light on the future with its contributions and values. As a result of the studies carried out on comparative education systems, countries have the opportunity to make changes to their systems by addressing the deficiencies of their own education systems. At this point, comparative education studies are seen as an essential field of study to get to know the education systems of different countries [48]. In this context, it is thought that comparing outdoor education practices in the preschool period between countries will fill the gap in the literature in this regard. The research to be carried out in this context will contribute to the literature in determining the similarities and differences

in education systems, as well as identifying the missing aspects, by comparing the existing preschool period outdoor education opportunities and activities in Finland, Ireland, Estonia, and Turkey.

The purpose of this research is to comparatively examine the existing pre-school education practices in Finland, Estonia, Ireland, and Turkey under the scope of 21st-century skill. In line with this purpose, answers are sought to the following research questions.

1. What are the specified aims of pre-school outdoor education environments in Finland, Estonia, Ireland, and Turkey?
2. What are the outdoor education environments identified at the preschool level in Finland, Estonia, Ireland, and Turkey?
3. What are the activities devised for outdoor education environments in Finland, Estonia, Ireland, and Turkey?
4. What are the preschool outdoor education modules used in Finland, Estonia, Ireland, and Turkey?
5. What are the assessment methods used in preschool outdoor activities in Finland, Estonia, Ireland, and Turkey?

## 2. Materials and Methods

The research model, data collection tools, research material, data analysis, and its steps are provided in this section.

### 2.1. Research Model

In the study, the first aim was to comparatively examine the outdoor education practices of Finland, Estonia, Ireland, and Turkey in terms of four variables. A comparative education approach was used to achieve this purpose. Comparative education is evaluated as a methodology and has various approaches such as horizontal, vertical, problem-solving, case study, descriptive, explanatory, and evaluative [49,50]. Horizontal and diagnostic approaches were jointly used in the study. While the horizontal approach is defined as an approach in which the dimensions in the education systems are handled one by one and all the variables belonging to that period are brought together to determine the differences [51]; in the descriptive approach, the relevant literature is examined for the topic in question and a comparative study of the similarities and differences are carried out [49]. In this study, the environments in which outdoor education practices are carried out in the early childhood period of selected countries were compared in terms of the aims of the education system, the modules used for outdoor practices, the activities used, and evaluation methods used.

### 2.2. Data Collection Process

Documents subject to document analysis have been selected with the purposive sampling method. The selected documents ought to be comprehensive and selective to carry out a holistic assessment [52]. Document analysis includes the analysis of written materials containing information about the facts targeted for research purpose [53]. The documents selected for document analysis were in line with the research purpose; current pre-school education programs in Finland, Estonia, Ireland, and Turkey were obtained from the respective official web pages of the Ministry of Education. These documents consist of the Turkish Pre-Primary Education Program [46], the Finnish Foundations of Early Childhood Education [54], the Estonian National Curriculum for Pre-School Children [55], and the Irish Pre-Primary Education Program [56,57].

### 2.3. Data Analyses Process

In this research, the process specified by Forster (1995) was taken into consideration for the document analysis process. According to Foster, document analysis is carried out according to the stages of (1) reaching the documents, (2) checking the originality, (3) understanding the documents, (4) analyzing the data, and (5) using the data [53]. In

this framework, (1) at the first stage, pre-school education programs, articles, and web addresses of the relevant countries are available and accessed through their websites.

The modules of Eurydice that explain the national education of the Finnish education system, the Estonian education system, and the Irish education system were accessed and downloaded to the researcher's computer. (2) The documents were accepted to be original since the related documents were obtained from the official web pages. (3) The themes related to the subject of this research were determined in advance by the researcher and 3 experts, and it was agreed that the information to be obtained according to these themes was in the relevant documents. (4) The collected data has been uploaded to the excel program on the computer and made ready for analysis. The descriptive analysis technique was used for the analysis. (5) As a result of descriptive analysis, the similarities and differences between the programs of each country were revealed, tabulated and interpreted.

Document analysis is used to systematically examine the content of written documents [58]. The descriptive research technique tries to reveal the events and facts as they are and ensures the data collection and data analysis occurs according to the previously determined conceptual framework and themes [59]. In this respect, the 4 themes determined according to the conceptual framework were: (1) objectives, (2) outdoor educational environments, (3) educational activities out of the classroom, (4) outdoor educational activity module.

### 2.4. Validity and Reliability Analysis

Lincoln & Guba indicate that, to ensure validity and reliability in qualitative research, the elements of credibility, transferability, consistency, and confirmability are necessary steps that must be performed [60]. In scientific research, credibility is an important aspect. Therefore, the long-term interaction and depth-oriented working process were used as the appropriate strategy to achieve credibility. The researchers' interaction for document analysis took 3 months. The researchers examined the obtained interview scripts in-depth, compared the concepts used in the scripts and compared them to each other. The two researchers worked together in the data analysis stages to ensure consistency and confirmability. During this process, the raw data obtained from the documents, and the codes obtained during the analysis from the source documents were classified under relevant themes. The findings and their interpretation were mutually examined for confirmation and consistency, and a consensus was reached. We also ensured the transferability process by clearly stating the findings obtained under the tables as direct quotations.

### 3. Results

In this section, the findings obtained from the research questions are presented in tables and interpreted.

The first sub-goal of the research was "What are the stated aims of pre-school education in Finland, Estonia, Ireland and Turkey?" Findings related to the question are shown in Table 1.

The examination of Table 1: Among the objectives stated in Finland's education program on outdoor education is "to provide children with positive learning experiences based on artistic and pedagogical activities based on play, movement, activity, seeing children as active recipients of information, raising children as members of society and ensuring the adaptation of the child to the natural environment, It is to create a play and educational environment, to ensure the development of children's personalities with other children through various activities, and to encourage the use of opportunities in the environment close to children." When the Estonian education program is examined, the goal of the stated objectives regarding the outdoor education environments is "to meet the child's need for movement, to prepare an environment for learning through play, to enable them to socialize and to support their learning by imitating, watching, discovering, experimenting, communicating". In the Irish education program, the objective is " to provide environments for children to play and socialize freely, to create environments

where they can use their imaginations and initiatives, and to create environments for the intellectual development of the child".

**Table 1.** Findings on the objectives of preschool education in Finland, Estonia, Ireland and Turkey.

| Country | Stated Objectives of Early Childhood Education Regarding Outdoor Educational Environments |
|---------|------------------------------------------------------------------------------------------|
| Finland | • To provide the child with a positive learning experience based on play, movement, artistic and pedagogical activities.<br>• To see children as active recipients of information,<br>• To ensure that children are brought up as members of society and adapted to the natural environment.<br>• To ensure the development of children's personalities with other children through various activities by creating a game and educational environment.<br>• To encourage children to use the possibilities of the immediate environment. |
| Estonia | • To meet the child's need for movement.<br>• To prepare an environment for learning through games.<br>• To support their socialization.<br>• To support learning by imitating, watching, discovering, experimenting, communicating, playing and exercising. |
| Ireland | • To provide environments where children can play and socialize freely.<br>• To provide environments where children can use their imagination and initiative.<br>• To prepare an environment for the child to have emotional, social, physical, creative, stimulating and intellectual development. |
| Turkey | • No objectives were determined regarding outdoor education activities in Turkey's pre-school education program. |

The second sub-goal of the research is "What are the outdoor education environments used in the pre-school period in Finland, Estonia, Ireland and Turkey?" Findings related to the question are shown in Table 2.

The examination of Table 2 suggests that the outdoor pre-school education environments used in Finland are "Nature, Gardens, Playgrounds and other constructed areas"; in Estonia "nature, natural environment, artificial environment and virtual environment" are used; in Ireland, "open spaces, sand and water areas, climbing areas, wheeled vehicles area, imagination area, creativity area, snack area, garden, horticultural areas and construction sites" are used; and in Turkey, environments such as "open space and garden" are mentioned along with field trips.

The third sub-goal of the research was: "What are the outdoor activities carried out in the pre-school education environments of Finland, Estonia, Ireland and Turkey?" Findings related to the question are provided in Table 3 below.

The study of Table 3 suggests that the activities performed outside the classroom during the pre-school period are "play, exercise, exploration, work, self-expression and artistic activities" in Finland; "play, physical activities, observation, exploration, artistic activities" in Estonia; "Unstructured play (free play), Semi-structured" play, Movement activity, field trip" in Turkey; and "Art and Design, Music, Drama and imaginary play, Language and Literacy, Mathematics and Numerical, Personal, Social and Emotional, Physical Development and Movement activities" in Ireland.

The fourth sub-goal of the research is: "Is there an outdoor education module in the pre-school education in Finland, Estonia, Ireland and Turkey? What are its contents?" Findings related to the question are presented below in Table 4.

**Table 2.** Outdoor education environments used in the pre-school period in Finland, Estonia, Ireland and Turkey.

| Country | Outdoor Education Environments |
|---|---|
| Finland | <ul><li>Nature</li><li>Gardens</li><li>Playgrounds</li><li>Other constructed areas</li></ul> |
| Estonia | <ul><li>Nature</li><li>Natural Habitat</li><li>Artificial environments</li><li>Digital Space</li></ul> |
| Ireland | <ul><li>Open spaces</li><li>Sand and water areas</li><li>Climbing areas</li><li>Wheeled vehicle area</li><li>Imagination space</li><li>Creativity (music/art/design) area</li><li>Snack area</li><li>Garden (different height)</li><li>Horticultural areas</li><li>Construction sites</li></ul> |
| Turkey | <ul><li>Open spaces</li><li>Garden</li><li>Field trips</li></ul> |

**Table 3.** Outdoor Educational Activities held in early childhood in Finland, Estonia, Ireland and Turkey.

| Country | Outdoor Educational Activities |
|---|---|
| Finland | <ul><li>Play</li><li>Exercise</li><li>Discovery</li><li>Work</li><li>Self-expression</li><li>Artistic activities</li></ul> |
| Estonia | <ul><li>Play</li><li>Physical activities</li><li>Observation</li><li>Discovery</li><li>Artistic activities</li></ul> |
| Ireland | <ul><li>Art and design</li><li>Music</li><li>Drama and imagination games</li><li>Language and Literacy</li><li>Mathematics and numerical activities</li><li>Social-emotional activities</li><li>Physical development and movement</li></ul> |
| Turkey | <ul><li>Unstructured gameplay</li><li>Semi-structured gameplay</li><li>Movement activity</li><li>Field trip</li></ul> |

**Table 4.** Outdoor education module in the pre-school education in Finland, Estonia, Ireland and Turkey.

| Country | Module Name | Content |
|---|---|---|
| Finland | I do research and work in my environment | • Math activities<br>• Science activities<br>• Environmental education events<br>• Daily life activities<br>• Activities to develop the concepts of change and time<br>• Activities to develop height, space perception and measurement skills<br>• Memory-building games and exercises with the potential to categorize, compare, sort, find and produce orders<br>• Observing and measuring,<br>• Activities to develop problem-solving skills,<br>• Activities to explore plants, animals and natural phenomena. |
| Estonia | Me and the Environment | • Awareness of the surrounding habitat through play and daily activities, from daily life and the surrounding social environment, including the natural and artificial environment,<br>• Activities to enable one to perceive the environment with different senses and through exploration and experience,<br>• Activities are done by observing, smelling, tasting, touching and listening to sounds,<br>• Comparing, modelling, measuring, calculating, speaking, reading, physical exercise, artistic and musical activities,<br>• Treating the environment with care and protecting the environment (plants, fungi and animals)<br>• Developing different time cycles, movement abilities, movement skills and other movement capacities (stamina, strength, speed, flexibility),<br>• Doing sport, developmental, and physical activities (cycling, skiing, skating, swimming). |
| Irelan | Outdoor Learning Book | • Activities to observe and experience outdoor features using all of their senses,<br>• Listening to various sounds, rhythms and songs in the outdoor environment, holding rhythms, instrument playing activities,<br>• Activities of playing cooperatively, negotiating roles, agreeing on rules,<br>• Activities for planning and designing scenarios, activating thoughts, ideas, emotions and imagination,<br>• Activities to strengthen self-confidence and oral language skills,<br>• Activities to develop confidence, self-esteem and sense of security and independence skills,<br>• Activities to comprehend the importance of physical activity for health and physical wellness, |
| Turkey | - | - |

The examination of Table 4 suggests that, within the Finnish preschool education program, "I Do Research and Work in My Environment Module" is a commonly used module; in Estonia, there is the "Me and the Environment" module; and in Ireland, there is the "Outdoor Learning Book" module. However, there is no module or outdoor learning book in the Turkish preschool education program.

Finland's "I do research and work in my environment" module includes "Mathematics, science, environment, daily life, time, space, plant, animal, natural life, observation, measurement, problem-solving potential to categorize, compare, sort, find and produce orders". In Estonia, 'Within the Self and Environment' module includes content such as "Memory development games and exercises with activities to enable the students to perceive the environment with different senses, activities done by observing, smelling, tasting, touching, listening to sounds, comparing, modelling, measuring, calculating, speaking, reading, physical exercise, artistic and musical activities, treating the environment with care and protecting the environment (plants, fungi and animals), different time cycles, movement abilities, movement skills and development of other movement capacities (endurance, strength, speed, flexibility), Sportive, developmental, and physical activities (cycling, skiing, skating, swimming)." Ireland's Outdoor learning book incorporates activities of observing and experiencing outdoor features using all senses, listening to various sounds, rhythms

and songs in the outdoor environment, rhythm keeping, instrument playing activities, playing cooperatively, negotiating roles, agreeing on rules, planning scenarios, designing, thoughts, activities to activate their ideas, feelings and imagination, activities to strengthen self-confidence and oral language skills, activities to develop confidence, self-esteem and sense of security and independence skills, activities to comprehend the importance of physical activity for health and vitality.

The fifth sub-goal of the research is: "What are the assessment methods for outdoor practices used in preschool education used in Finland, Estonia, Ireland and Turkey?" The related findings are presented in Table 5.

**Table 5.** The assessment methods for outdoor practices used in preschool education used in Finland, Estonia, Ireland, and Turkey.

| Country | Assessment Methods |
|---------|-------------------|
| Finland | • Evaluation of the child (observation, monitoring, measuring level states)<br>• Assessment by teacher |
| Estonia | • Evaluation of the child's development<br>• Observation (in-class, free play, Structured play)<br>• Parent cooperation |
| Ireland | • Evaluation of students<br>• Observation (observation, evaluation and recording)<br>• Teacher-teacher collaboration<br>• Teacher's end-of-day evaluation<br>• Parent cooperation |
| Turkey | • Conversation (activity evaluation with big groups)<br>• Observation |

The examination of Table 5 points out that the assessment methods of outdoor activities are: "evaluation of the child (observation, monitoring, measuring level status) and teacher evaluation" in Finland; and the assessment methods in Estonia are "evaluation of the child's development, observation and parent cooperation". In Ireland, the methods used are "students' evaluation, observation, teacher-teacher cooperation, teacher's end-of-day evaluation and parent cooperation", whereas the methods of assessment are "conversation and observation" in Turkey.

## 4. Discussion, Conclusions and Results

In this study, the aim was to compare the out-of-class educational activities in the preschool stages of Finland, Estonia, and Ireland, which are members of the OECD, with the out-of-class educational activities of the preschool education carried out in Turkey.

When the aims of the programs regarding outdoor educational activities in early childhood in Finland, Estonia, Ireland, and Turkey are examined, there are no listed objectives for outdoor education activities in the Turkish preschool education program, whilst the stated objectives of meeting the child's need for play and movement are a common goal in the pre-school education program of Finland, Estonia, and Ireland. In early childhood education, it is important to create opportunities for the development of the child, including both indoor and outdoor learning, and to provide various creative activities that provide good socialization conditions for children [61]. It has been concluded that children have common goals such as being active, experimenting, communicating, and gaining basic skills such as discovery. Play, observation, research, questioning, experimentation, and being active in different environments have shown positive effects on the development of motor skills, health and, concentration in young children [62–65].

The comparison of the early childhood outdoor education environments of Finland, Estonia, Ireland, and Turkey were examined, and it was determined that open space

nature and school gardens were common environments, while the country with the most diverse outdoor education environment was Ireland. The country with the least variety was Turkey. The study of relevant literature pointed out that the effect and permanence of out-of-class learning environments on lifelong learning skills affect students' participation in out-of-class learning environments and their participation in educational activities in their professional lives [41]. Educational environments organized for child development should appeal to children's play, physical activity, exploration, artistic experiences, natural curiosity, and learning desires [66].

The comparison of the early childhood extracurricular activities of Finland, Estonia, Ireland, and Turkey pointed out that the common activity of all countries is for students to play and perform physical movement activities. Outdoor activities in Finland include exploration, work, self-expression, and artistic activities in addition to play and exercise. In comparative observation, discovery and artistic activities, in addition to playing and physical movement, are important in Estonia. The study pointed out that the activities carried out in Finland and Estonia are similar. In contrast, the study showed that there are numerous activities planned in Ireland such as art and design, music, drama, language and literacy, mathematics and social and emotional activities, in addition to play and physical movement activities. In contrast, the outdoor activities carried out in Turkey were play and movement activities. The Estonian national curriculum for preschool childcare institutions (2008) emphasizes the importance of working through play, observation, and research in different settings. In this regard, the importance of extracurricular activities comes into focus. In previous studies, out-of-class activities, outdoor games, and their contribution to child development were mentioned. According to [67], successful outdoor activities offer flexible opportunities where children engage in creative, imaginative play; develop communication skills; and build relationships with other children and adults. The time and space allocated for children to play outdoors is recognized as both a need and a right and is at the center of their well-being and development [68].

The comparison of Finland, Estonia, Ireland, and Turkey's early childhood extracurricular activities modules suggests that Finland's pre-school education curriculum includes the "I Do Research and Work in My Environment" module, observation, problem-solving, discovery activities, and these occurred outside the classroom. It shows that outdoor education activities are carried out depending on the activities included in this module, which includes the "Me and the Environment" module in the Estonian preschool education program. It has been determined that there is an "Outdoor Learning" module associated with the Irish preschool education program, and various activities are carried out within the framework of this module. The fact that Turkey does not have a module that hosts any outdoor education activities is among the striking findings of the research. The countries have training modules in addition to their training programs; it facilitates making the goals desired to be achieved and the qualities desired in the individual being made in a planned and programmed way. Modules or guides that support learning activities, in line with the interests, needs, and abilities of the student, guide the teacher while directing the teaching by putting the student into the center [69].

The assessment methods used for early childhood extracurricular activities in Finland, Estonia, Ireland, and Turkey. The findings suggest that all countries use observation to evaluate their students in the evaluation process. In Estonia and Ireland, the teachers are directed to cooperate with parents in addition to student evaluation. Similar processes also apply to Finland and Ireland. In addition, teacher evaluation is among the findings of this research. "Single-type" assessment in early childhood will not be sufficient to identify and meet the needs of the child in this period [70]. The evaluation process, first of all, the skill or skills to be evaluated should be determined and then the most appropriate method or methods should be selected for evaluation.

In the light of these results, we recommend the development of a teaching module and book for pre-school pedagogy in Turkey, one that includes out-of-class education activities in the preschool education process as in Finland, Estonia, and Ireland. Teachers

will be able to carry out the teaching activities to be performed outside in accordance to the plans and determined goals due to the developed modules and books for outdoor teaching. It is an advantage for the teachers if the activities are written in a module or a booklet. In addition, it has also been noted that inclusive activities, especially ones that raise environmental and nature awareness, have a separate importance and priority in terms of current global problems. It is effective to raise this awareness with outdoor activities; therefore, such activities should definitely be added in the outdoor education process. In particular, evaluation gains a special importance, and is difficult in outdoor education activities. The specified assessment methods in the module or in an activity book will better guide the teachers. It is accepted that children should not be subjected to a uniform assessment in this period.

**Author Contributions:** Conceptualization, K.Ö. and U.A.; methodology, K.Ö. and U.A.; software, K.Ö.; validation, K.Ö. and U.A.; formal analysis, K.Ö. and U.A.; investigation, K.Ö.; resources, K.Ö. and U.A.; data curation, K.Ö. and U.A.; writing—original draft preparation, K.Ö. and U.A.; writing—review and editing, K.Ö. and U.A.; visualization, K.Ö.; supervision, U.A.; project administration, U.A. All authors have read and agreed to the published version of the manuscript.

**Funding:** This research received no external funding.

**Institutional Review Board Statement:** Not applicable.

**Informed Consent Statement:** Not applicable.

**Data Availability Statement:** Data can be requested from the researchers.

**Conflicts of Interest:** The authors declare no conflict of interest.

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
