# Peer review of "An Investigation of Preschool Level Out-of-Class Education Activities in Finland, Estonia, Ireland, and Turkey within the Framework of 21st Century Skills"

_sustainability, doi:10.3390/su14148736_

Round 1
Reviewer 1 Report
The study provides a comparative examination of the curricular conditions for outdoor learning activities in early childhood in a set of countries. While the study provide a good layout of this comparison, some issues need to be adressed before the study can be ready for publication.
For one, the rationale for learning activities can be more conceptually explored. Why is this examination undertaken? One interesting commonality can be seen in the current trend on children’s active learning, that is combining activities such as outdoor learning with learning activities for children. See Science recent issue on this:
Yannier, N., Hudson, S. E., Koedinger, K. R., Hirsh-Pasek, K., Golinkoff, R. M., Munakata, Y., Doebel, S., Schwartz, D. L., Deslauriers, L., McCarty, L., Callaghan, K., Theobald, E. J., Freeman, S., Cooper, K. M., & Brownell, S. E. (2021). Active learning: “Hands-on” meets “minds-on”. Science, 374(6563), 26–30. https://doi.org/10.1126/science.abj9957
In this literature, guided play and playful learning activities (including outdoor activity and others) have been related to skills such as reading, see for example: Toub, T. S., Hassinger-Das, B., Nesbitt, K.T., Ilgaz, H., Weisberg, D. S., Hirsh-Pasek, K., …, and Dickinson, D. (2018). The language of play: Developing preschool vocabulary through play following shared book-reading. Early Childhood Research Quarterly. 45, 1–7. Doi: 10.1016/j.ecresq.2018.01.010
This can be a move to further express why this study is undertaken, and to show its relevance in an international journal.
As for now it seems as the study is aimed to single-out the lack of outdoor activity in Turkey. While curricular advice may be a by-product of the study - it cannot - as it stands make claims of curricular reform - as the study does not clearly show why outdoor learning ought to be a part of curricula. If this recommendation is to be made, it should be better conceptualised and more nuanced, taking in differences in cultural context and local possibilities of outdoor play and learning activities (for example a large geographical difference between urban and rural areas might disallow this to be specified in a national curriculum).
Some specific comments to the text:
p2. l. 96-97 ”theogainntury” should be 21st century?
p3 l. 100 ". It is an undeniable fact” - perhaps this phrase should be avoided in scientific writing
p3 l. 144 "and is the period when child development is completed.”. If child development is ”completed”, it is not definitely at age 6. This should be rewritten. Do the authors means that a certain period of child development is ”completed” at age 6, if so - specify.
p.4 l 160 "35] comparative education” This paragraph starts with a reference and there seem to be an error or
Misc. comments: There are some spelling and grammatical errors throughout the paper. It should be carefully read and fixed during revision.
Author Response
We thank the reviewers and Editor for valuable comments and suggestions. The paper has been revised, considering all the reviewers' comments and suggestions. The reviewer response file is attached.

Reviewer 2 Report
I would recommend a terminologically unified approach, currently the title is "Outdoor Education", the abstract "out-of-class education", "extra-class activities"
As for the countries, reasons have been presented as to why Estonia, Finland and Ireland have been chosen, but it remains somewhat unclear why Turkey has been chosen.
- As for the countries, reasons have been presented as to why Estonia, Finland and Ireland have been chosen, but it remains somewhat unclear why Turkey has been chosen. The justification given is that these countries are in first place in the PISA ranking. Perhaps this rationale could be considered here, since the PISA study is conducted among elementary school students, but not in preschools, which is the focus of this article.
I would expect a little more overview of previous outdoor education research results in the introduction, perhaps the introduction could be structured accordingly.
Could there be a few sentences about each country's preschool system?
Line 394 is for some reason given as a reason for the countries that Estonia, Finland, Iceland are members of the OECD, but this is perhaps not related to the implementation of outdoor education
Could the results highlight the focus on child development and the integration of learning areas, which is characteristic of Estonia, Finland and Ireland?
In theory, great emphasis is placed on the skills of the 21st century, but in the didcussion it is no longer clear how and if different countries support the skills of the 21st century?
The design of Table 4 should be revised.
Line 245 is an incomplete sentence.
It should be checked that all added links open correctly, currently for example the web pages mentioned in line 494 do not open
Would be a web page, it is not otherwise published material - Halligan, M. W. (2006). Outdoor education for middle school youth: A grant proposal project. California State University
It is mentioned in line 214 that the programs of all countries were obtained from the official web pages of the Ministry of Education. all the web addresses where the programs are taken should be included, they are not in the sources, line 566 does not have a source name, only a web address.
There are many errors in the formatting of sources, they should be corrected.
Author Response

(The authors gave the same response as above.)

Round 2
Reviewer 2 Report
The article has been improved and corrected according to my recommendations.